# How Gut Microbes Nurture Intestinal Stem Cells: A *Drosophila* Perspective

**DOI:** 10.3390/metabo12020169

**Published:** 2022-02-10

**Authors:** Constantina Neophytou, Chrysoula Pitsouli

**Affiliations:** Department of Biological Sciences, University of Cyprus, 1 University Avenue, Aglantzia, Nicosia 2109, Cyprus; kneofy02@ucy.ac.cy

**Keywords:** intestinal stem cells, midgut, microbiota, nutrients, vitamins, fatty acids, cholesterol, amino acids, sugars

## Abstract

Host-microbiota interactions are key modulators of host physiology and behavior. Accumulating evidence suggests that the complex interplay between microbiota, diet and the intestine controls host health. Great emphasis has been given on how gut microbes have evolved to harvest energy from the diet to control energy balance, host metabolism and fitness. In addition, many metabolites essential for intestinal homeostasis are mainly derived from gut microbiota and can alleviate nutritional imbalances. However, due to the high complexity of the system, the molecular mechanisms that control host-microbiota mutualism, as well as whether and how microbiota affects host intestinal stem cells (ISCs) remain elusive. *Drosophila* encompasses a low complexity intestinal microbiome and has recently emerged as a system that might uncover evolutionarily conserved mechanisms of microbiota-derived nutrient ISC regulation. Here, we review recent studies using the *Drosophila* model that directly link microbiota-derived metabolites and ISC function. This research field provides exciting perspectives for putative future treatments of ISC-related diseases based on monitoring and manipulating intestinal microbiota.

## 1. Introduction

The physiological inhabitants of the intestine, the gut microbiota, include bacteria, archaea, viruses and fungi that have coevolved with their hosts. The intestine offers an excellent mucosal environment for reciprocal host–microbe interactions and harbors a vast number of microorganisms, which are separated from the internal milieu by a single epithelial cell layer [1,2]. The symbiotic relationship between the host and the microbiota is mutually beneficial and allows the host to perform various necessary metabolic functions, including nutrient absorption and energy regulation [3]. A couple of decades ago, the gut microbiota was characterized as a neglected virtual organ within an organ due to its specific location, endocrine action and metabolic activity [4]. The composition of the commensal bacterial communities has been the focus of numerous studies in recent years. Metagenomic analyses have greatly facilitated the characterization and underscored the complexity of the intestinal microbiota. The human gut microbiome encompasses approximately 1000 different microbial species [5], many of which remain uncultivated. Interestingly, gene expression studies have revealed an enrichment in genes encoding metabolic functions in microbiota compared to the human genome [6].

The host-gut microbiome interactions are central modulators of different aspects of host physiology, including growth, development, nutritional status, behavior and immunity [1,7] (Figure 1). The microbiota directly affects intestinal function through its contribution to energy harvest and storage, micronutrient synthesis, including that of vitamins that the host body is unable to synthesize, enhancement of fermentation-mediated digestive efficiency and absorption of undigested nutrients [8]. In addition, the microbiome plays a key role in the physiological development and training of the immune system [9], as well as the local induction of immune responses that regulate commensals and pathogens [10]. It also stimulates the activity of intestinal stem cells (ISCs) [11,12] and contributes to homeostasis maintenance [13]. In turn, the resident gut microbes absorb energy and metabolites from the host and its diet to support their growth [14]. This mutually beneficial interaction between the microbes and the host is critical for organismal health.

Contrariwise, imbalance in microbiota composition or abundance leads to microbial dysbiosis, which is associated with autoimmune and allergic diseases, diabetes, obesity and inflammatory bowel disease [7]. Increasing evidence indicates that the intestinal microbiota is also an important contributor to colorectal carcinogenesis, progression and metastasis [15]. Indeed, dysbiosis-associated intestinal inflammation has been associated with colorectal cancer in both mice and humans [16,17]. Pathogenic bacteria, as well as pathobionts (namely physiologically symbiotic species exhibiting abnormal overgrowth and activity that turns them into pathogens) may induce a tumorigenic environment by secretion of mediators, such as interleukins (e.g., IL-6) and reactive oxygen species (ROS), which subsequently drive epithelial cell proliferation [18]. Since it is widely believed that the ‘cells-of-origin’ of different types of cancer are tissue-specific stem cells, there is an evolving number of studies that aim to characterize the molecular factors linking intestinal microbiota to host metabolism to ISC activity.

In this minireview we provide a *Drosophila* perspective on the effects of dietary interventions on the composition of bacterial communities within the gut and the impact of commensals on metabolism, with a special focus on the effects of microbe-derived nutrients on ISC behavior and epithelial renewal.

## 2. The *Drosophila* Midgut Environment

The genetically powerful invertebrate model *Drosophila melanogaster* has contributed fundamental insights into our understanding of processes governing multicellular organism development, physiology, homeostasis, signaling pathways and aging [19]. The *Drosophila* digestive tract, in particular, resembles the gastrointestinal tract of mammals at the structural, functional and cellular levels [20]. Importantly, the middle part of the *Drosophila* intestine, the midgut, corresponds to the mammalian small intestine and encompasses ISCs, which, similar to their mammalian counterparts, are self-renewing, multipotent cells, able to generate all the differentiated gut cell types. In the *Drosophila* midgut, the ISCs are distributed throughout the basal side of the pseudo-stratified epithelial monolayer [21,22]. In mammals, the small intestine presents a characteristic single-layered epithelial morphology encompassing crypts and villi [23]. ISCs are located at the crypts away from the intestinal content, whereas differentiated gut cells populate the villi, which directly contact the intestinal lumen [24]. ISCs are the cellular source of all mature cell types of the intestinal epithelium during adult life. In *Drosophila*, ISCs transiently differentiate to either enteroblasts (EBs) or pre-enteroendocrine cells (pre-EEs), which, in turn, terminally differentiate into absorptive enterocytes (ECs) or secretory enteroendocrine cells (EEs), respectively [21,25,26]. Specialized differentiated cells, such us antimicrobial-secreting Paneth cells, mucus-secreting goblet cells and mechanosensing tuft cells, are absent from the fly midgut lineage [27,28]. The *Drosophila* midgut epithelial tube is ensheathed by longitudinal and circular visceral muscle fibers involved in peristalsis, is oxygenated by visceral tracheae, and is innervated by neuronal termini at the anterior and posterior ends [21,29,30,31,32].

The *Drosophila* midgut, similar to the mammalian intestine, acts as a barrier to ingested environmental fluids and microorganisms and, at the same time, allows for nutrient absorption. To achieve these functions, the polarized midgut ECs encompass apical occluding or septate junctions above the lateral adherens junctions, contrary to other *Drosophila* epithelia, whereby the junctional complexes are reversed. This junctional arrangement, which is similar to mammalian epithelia, may have evolved to prevent luminal contents and intestinal microbiota to access the lateral sides of the cells [33]. The adult fly midgut is structurally and molecularly compartmentalized along the anterior-posterior axis, so that different parts of the organ perform sequentially different functions [34,35]. Specifically, the *Drosophila* midgut is morphologically divided into 5 broad domains (R1-R5), which are further subdivided into 14 regions exhibiting differential gene expression [35]. The fly midgut also interacts with distant tissues via hormonal secretion by specialized EE cells [36,37].

In addition to the cellular simplicity observed in the fly midgut, *Drosophila* can also benefit microbial research by deconstructing the complex multi-microbial interactions in vivo. Contrary to the mouse and human gut, which are populated by hundreds of microbial species, the *Drosophila* intestine is populated by a simple microbial flora composed by less than 20 culturable bacterial species [38]. The gold standard approach for establishing causality of a single pathogen on an associated condition uses germ-free animals. Although mice can be treated to become germ-free, major limitations, including cost, duration of the studies, technical challenges, infrastructure demands and small experimental sample size, highlight the need for an alternative model organism [39]. In *Drosophila,* it is easier to generate and manipulate germ-free flies, as well as gnotobiotic individuals, which are germ-free animals re-associated with selected microbes. The simplicity and intrinsic-tractability of the *Drosophila* intestinal microbiome, as well as its inexpensive maintenance and fast life cycle, facilitate the molecular characterization of the mutualistic host–microbiota relationships [39,40], which is possibly conserved in mammals [41].

Even though the fly lacks an adaptive immune system, it has been instrumental in the identification of innate immunity mechanisms and the characterization of the role of conserved signaling pathways (e.g., the NF-κB pathway) in this process [42]. Microbiota interacts with intestinal epithelial cells and induces innate immune responses via the NF-κB/IMD pathway and NADPH oxidase-mediated reactive oxygen species (ROS) production. Symbiotic bacteria induce the NF-κB/IMD pathway via peptidoglycan, and the NADPH oxidase Nox via lactate [12,43], whereas pathobionts and pathogens produce uracil that induces the NADPH oxidase Duox [44,45]. Intestinal homeostasis and regeneration upon bacterial or chemical damage is mediated by ISC mitosis and differentiation, which are, in turn, controlled by conserved cell communication pathways, including the epidermal growth factor receptor (EGFR), the Jak/STAT, the Wnt/Wingless (Wg), the insulin/insulin-like growth factor-1 (IGF-1) signaling (IIS) and the target of rapamycin (TOR) pathway (Figure 2).

## 3. *Drosophila* Nutrition Shapes Microbiota Composition

Similar to mammals, the diet is considered among the most significant determinants of interindividual microbiome variability in *Drosophila*. Microbiota composition is altered significantly upon shifts from plant-based to animal-based diets, as well as by alterations in protein and carbohydrate intake levels [46,47,48]. There are more than 70 different published diets used for microbiome research in *Drosophila* that vary significantly in the type and number of components used [49]. The microbiome of both laboratory-raised and wild-caught flies is enriched in the *Firmicutes* phylum, represented by the Lactobacillaceae and Enterococcacea families, and the two groups (*alpha* and *gamma)* of the *Proteobacteria* phylum, represented by the Acetobacteraceae and Enterobacteriaceae families. *Acetobacter pomorum, A. tropicalis, Lactobacillus brevis, L. fructivorans and L. plantarum* are among the most abundant species [50]. Less abundant species, identified as opportunistic pathogens, include *Providencia, Serratia, Erwinia, Pantoea* and *Pseudomonas* [51]. Diets rich in complex polysaccharides, such as cornmeal and soy flour, favor the growth of *Lactobacillus* species, whereas sugar-rich diets increase abundance of *Acetobacteraceae*, and particularly the *Acetobacter* and *Gluconobacter* species. In addition, increased yeast intake steeply induces the abundance of the predominant species [52].

Strikingly, intestinal microbes can substitute dietary components in nutrient-scarce conditions. For example, *Issatchenkia orientalis,* a fungus isolated from wild-caught flies, promotes dietary amino acid harvest and rescues lifespan of flies feeding on protein-deficient diets [53]. Moreover, upon nutrient scarcity, *L. plantarum* is sufficient to promote growth by regulating hormonal signals via the TOR-dependent host nutrient sensing system [54]. In addition, continuous administration of heat-killed microbes (fungi and bacteria) to flies has been reported to rescue protein undernutrition phenotypes, suggesting that dead microbes function as protein sources [55]. Under nutrient-poor conditions, bacterial species can be readily stimulated to share metabolites and consequently sustain each other’s biosynthetic capabilities [56,57,58].

During development, nutritional needs change constantly due to the different energy demands of the growing animal. For example, *Drosophila* larvae need a protein-rich diet to support their fast growth, whereas the adult flies feed less to maintain energy homeostasis, and young female flies require more energy for egg production [59]. In addition, early intestinal maturation, growth and homeostasis of newly emerged adult flies depend on food intake [60]. Specifically, optimal food intake of young adult flies regulates the ISC pool by activating ISC symmetric divisions, controlled by the IIS pathway [60]. Similar to the variations in nutritional needs, the microbiota quantity and composition evolves during the *Drosophila* life cycle, with dominant fluctuations between *Lactobacillacea* and *Acetobacteraceae* [38]. Strikingly, several studies have shown that the microbiota has an impact on *Drosophila* lifespan. Nevertheless, whether the microbiota-lifespan link is positive or negative remains contradictory. For example, although the presence of gut bacteria during the first week of the fly’s adult life impinges positively on longevity [61], a number of studies have shown that elimination of microbiota extends fly lifespan [62,63,64,65,66], and that microbial abundance, but not diversity, is a significant determinant of *Drosophila* longevity [67]. Lifespan extension is favored under low nutrient conditions, which upregulate *dMyc* levels in ECs, maintaining their fitness and ISC proliferation, thus, improving gut homeostasis [68]. Taken together, these data underscore an indirect effect of microbiota on intestinal development, ISC-mediated intestinal maintenance and overall fly fitness.

## 4. Host Intestinal Regeneration-Microbe Interactions and Nutrition

The gut epithelium is a highly plastic tissue. It responds quickly to nutrient scarcity by reducing its volume via EC apoptosis and attenuation of ISC divisions, whereas food intake triggers expansion of progenitor cells and subsequent tissue growth [60,69]. Dietary stimuli mainly induce the IIS and the TOR pathways, leading to ISC proliferation and tissue growth [60,69,70,71,72,73]. The intestinal epithelium is directly exposed to diet- and microbiota-derived nutrients that control its homeostasis. Investigations on how dietary nutrients and dietary states influence ISCs have been reviewed recently [74], but the critical microbiota-derived nutrients that modulate ISC self-renewal and proliferation are only beginning to be recognized. Here, we will present an overview of the key microbiota-produced metabolites that directly or indirectly impinge on ISC biology focusing on the fly model (Figure 3).

### 4.1. Amino Acids

Dietary amino acids, as the building blocks of proteins, are crucial for the fitness of all animals. Most animals are unable to produce essential amino acids and, therefore, must acquire them from their environment: their diet or their microbiota [75]. An interesting intestinal interbacterial interaction beneficial for the fly host has recently been documented and involves two species of the bacterial microflora. Specifically, the very abundant fly intestinal bacterium *A. pomorum* produces the essential amino acid isoleucine (Ile). In turn, *A. pomorum*-produced Ile promotes *L. plantarum* growth in Ile-free media. Growing *L. plantarum* produces lactate that is essential for *A. pomorum* growth. Lactate production is also responsible for suppression of host protein appetite, in the presence of *A. pomorum*, highlighting the syntrophic interaction between these two species [57]. In an environment with low amino acids (such as the holidic diet) the *A. pomorum* supports the growth of *L. plantarum* by providing not only Ile, but also other essential amino acids for which *L. plantarum* is an auxotroph, such as arginine, leucine, valine and cysteine [76]. While this mutualistic interaction of bacteria is beneficial to the host in conditions of nutrient scarcity and supports reproduction [57], there is no evidence whether host ISC activity is affected. Nevertheless, other studies have demonstrated both autonomous and non-autonomous ISC activity regulation by dietary amino acids [77,78,79].

Particularly, depletion of the essential amino acid, methionine, and the methionine-derived S-adenosylmethionine, reduces midgut mitosis in *Drosophila* by controlling protein synthesis autonomously in ISCs and induction of the Jak/STAT ligand Unpaired 3 (Upd3) non-autonomously in ECs [78]. Moreover, diet supplementation with the conditionally essential amino acid glutamine exhibited ISC activating characteristics, including an increase in the total intestinal cell number, and more specifically the EEs [79]. Glutamine, as a substrate molecule for the rate-limiting enzyme glutamine fructose-6-phosphate aminotransferase (Gfat) [80], is used to control the nutrient-responsive hexosamine biosynthetic pathway (HBP) [81]. Gfat2, which is expressed in ISCs [82], is a critical gatekeeper of nutrient-induced ISC activation. In Gfat2 mutant ISCs, dietary supplementation with the HBP intermediate N-acetyl-D-glucosamine (GlcNAc) is sufficient to maintain ISC proliferation during caloric restriction independent of food intake [83]. In addition, dietary glutamate was found to stimulate ISC proliferation and growth via calcium signaling [77]. In flies, amino acid sensing is mediated by the TOR pathway [84,85,86,87,88]. The TOR pathway controls systemic growth by circulating hormones, Ecdysone and the *Drosophila* insulin-like peptides (dILPs) [89], and maintains ISC activity via the Jak/STAT, EGFR and Jun kinase (JNK) signaling pathways [90]. Both microbiota and monoassociation of germ-free flies with *L. plantarum* (*L. plantarum* gnotobiotic model) can substitute dietary amino acids during nutrient deprivation conditions. Nonetheless, genetic modulation of the TOR pathway cancels the beneficial impact of the bacteria, indicating that the host nutrient-sensing system acts genetically downstream of the microbiota [54].

### 4.2. Sugars

Sugar is the major energy source for most cells and, due to its high caloric content, supports essential organismal functions. However, a sugar-rich diet is associated with obesity [91] and chronic metabolic diseases, including hypertension and insulin resistance [92]. A *Drosophila* Type 2 diabetes model, whereby flies feed on increased sucrose concentrations, revealed that high sugar disrupts epithelial homeostasis by altering signaling pathways of the gut, as well as microbiota composition [93]. Specifically, excess sugar causes intestinal epithelial stress evident by the accumulation of ROS. Moreover, induction of the JNK pathway causes increased differentiation of ISCs, whereas ISC proliferation remains unchanged, despite the Jak/STAT pathway downregulation. Under these feeding conditions, the microbiota exhibit increased complexity, but reduced abundance [93]. However, whether changes in microbiota are causative for the observed phenotypes remains to be addressed. Further, sugar catabolism bacterial genes are found to benefit the host by reducing lipid content [94,95]. Specifically, flies associated with *A. tropicalis* exhibit increased consumption of dietary glucose, and reduced triglyceride content, which might be advantageous to the host during malnutrition [95].

### 4.3. Fatty Acids

Similar to high-sugar diet, high-fat diet (HFD) is strongly linked to obesity and obesity-related diseases [96]. HFD induces changes in intestinal structure and function [97,98], mediated mainly by alterations in ISC activity control. The mechanisms through which pro-obesity diets control ISCs and their function are not well characterized. A previous study in mammals has shown that specific fatty acids, including palmitic acid and oleic acid, directly interact with the ISCs. Fatty acids induce the peroxisome proliferator-activated receptor delta (PPAR-δ) specifically in ISCs and progenitor cells to enhance their stemness [99]. However, a more recent study has shown that a HFD containing triglycerides and saturated fatty acids induces a microbiota-dependent indirect transient activation of the ISCs [100]. This dietary regime alters both the abundance and the composition of microbiota leading to enrichment in *Enterobacteriales* and *Caulobacterales*, possibly due to the high energy gut content that benefits their growth. Mechanistically, HFD-induced stress leads to JNK pathway activation in ECs, secretion of the Upd3 ligand and activation of ISC proliferation through the Jak/STAT signaling pathway. Subsequent increase in the number of EEs, correlates with previous findings supporting that HFD alters the expression and release of gut hormones [101]. Juvenile hormone (JH) is an example of an endocrine signal that directly activates ISC proliferation for adaptation to size changes after mating, and induces the expression of genes involved in fatty acid synthesis, specifically in ECs [102]. In mammals, HFD-induced circulating hormones lead to reduced appetite that prevents overnutrition [103].

### 4.4. Short Chain Fatty Acids

Prebiotics are non-digestible fermentable food ingredients, which, upon ingestion, promote the growth of beneficial bacterial species already present in the intestine. Fermentation of prebiotics by intestinal microbiota directly produces beneficial end products, such as short chain fatty acids (SCFAs) [104]. SCFAs are fatty acids encompassing fewer than six carbon atoms (e.g., acetate, propionate, butyrate) [105]. SCFAs are crucial for intestinal cells, acting as a major energy source [106]; they also mediate whole-body metabolic functions and physiology [107].

In *Drosophila*, microbiota-derived SCFAs have been shown to [107] regulate lipid and carbohydrate metabolism to maintain ISCs. This process is mediated through recognition of SCFAs from EEs and secretion of small EE peptides [108,109]. For example, the microbiota-derived acetate has been shown to act via the membrane receptor PGRP-LC as an activator of the EE IMD pathway, leading to phosphorylation of the transcription factor Relish. IMD activation leads to secretion of the EE peptide Tachykinin (Tk), which, in turn, has been shown to regulate lipid metabolism and insulin pathway activation [110]. Another example of host-microbiota cooperation to sustain the gut-microbe mutualism comes from a study showing that the activity of the enzyme pyrroloquinoline quinone-dependent alcohol dehydrogenase (PQQ-ADH), which is responsible for acetic acid biosynthesis in *A. pomorum*, modulates the IIS pathway in flies. Specifically, it controls growth and metabolic homeostasis, ensures maintenance of basal ISC numbers and epithelial cell turn over via the JNK-mediated Jak/STAT pathway induction [111].

Apart from acetate, the SCFA butyrate has key roles in intestinal epithelial homeostasis in both mammals and flies. In mammals, butyrate stimulates growth and proliferation of epithelial cells, reduces inflammation and oxidative stress, improves barrier function and inhibits colon cancer development [112]. Interestingly, in the healthy intestine, ECs consume and metabolize butyrate, thus, preventing it from reaching the ISCs located at the crypts. Epithelial injury causing exposure of crypt ISCs to butyrate suppresses ISC proliferation and mucosal wound healing via Foxo3 regulation [113]. In *Drosophila,* the addition of sodium butyrate in the diet can effectively extend viability and life span [114] and microbiota-derived sodium butyrate impacts the microbiota composition and host gene expression [115]. Sodium butyrate was found to induce Firmicutes and to suppress Proteobacteria, while at the gene expression level, it was shown to have an inhibitory effect in key biological processes such as cell proliferation [115].

### 4.5. Cholesterol

Cholesterol is a key molecule involved in various physiological host functions, while its interaction with intestinal bacteria is known for decades. Nonetheless, the significance of this interaction in host health, and whether gut microbiota modulates cholesterol to benefit the host, has only started to be studied recently [116,117]. Microbiota is able to chemically modify cholesterol, which in turn affects microbial composition. Consequently, the signaling mechanisms they mediate are also affected [118].

In *Drosophila*, high cholesterol diets alter ISC cell differentiation by modulating the Delta ligand and Notch stability in the endoplasmic reticulum [97]. This direct interaction of cholesterol with the Notch signaling pathway causes a preferential differentiation of ISCs into EEs. Importantly, alterations in cholesterol concentration regulate susceptibility to EE tumors [97]. In addition, cholesterol is the precursor molecule for the steroid hormone 20-hydroxy-ecdysone (20HE) biosynthesis [119,120]. The 20HE stimulates egg production in female flies and acts on the midgut progenitor cells promoting EB differentiation to ECs through its effector *Ecdysone-induced-protein-75B (Eip75B)* to adapt midgut homeostasis [121]. The impact of microbiota on these observations needs to be determined.

### 4.6. Vitamins

Vitamins are essential micronutrients ensuring normal development and organismal health due to their involvement in vital cellular processes, such as fat and carbohydrate metabolism, as well as DNA synthesis [122]. Since mammals and *Drosophila* are unable to synthesize essential vitamins, they must acquire them from their diet or from enteric bacteria. Microbiota-derived vitamins can shape the intestinal microbial community and prevent dysbiosis. As not every bacterial strain can synthesize all different vitamins, some symbiotic species possess complementary biosynthetic pathways to maintain their growth [123].

Previously, microbiota was found to supplement *Drosophila* with the dietary B vitamins and sustain their growth in low-nutrient diets [124]. A recent study has shown that intestinal microbiota can support larval growth and development by supplementation of thiamine (Vitamin B1). Specifically, although germ-free larvae were unable to develop on a thiamine-free diet, gnotobiotic studies with *A. pomorum*, managed to rescue their development [125]. Moreover, metabolic cooperation between *Acetobacter* species and *L. plantarum* including, not only supplementation of amino acids, but also B vitamins, such as biotin and pantothenate, when flies are reared on holidic diets, supports growth of the two species that, in turn, supports host systemic growth [76]. Additionally, cell-specific transcriptome mapping of the adult *Drosophila* midgut has revealed that the biotin (Vitamin B7) sodium multivitamin transporter (Smvt) is specifically expressed in ISCs and EBs, highlighting the metabolic requirements of ISCs for this vitamin. Smvt has been found to be essential for ISC maintenance and tissue homeostasis, as well as for fly survival [82]. In a recent study, biotin transported to the ISCs via Smvt was found to be necessary for mitosis and physiological intestinal cell differentiation, in a process parallel to the Jak/STAT pathway. Furthermore, ISC-specific *Smvt* silencing leads to dysbiosis caused by increased load of the opportunistic pathogen *Providencia sneebia*. Strikingly, in biotin-scarce conditions, microbiota-produced biotin could directly induce ISC mitosis [126]. Taken together, these studies highlight an important role of bacterially-derived B vitamins on host physiology, immunity and ISC activity. However, the mechanisms of direct interaction between microbial vitamins and ISCs remain unclear.

### 4.7. Other Metabolites

In previous studies, intestinal dysbiosis due to reduced abundance of *Lactobacillus spp*., specifically *L. plantarum*, and increased abundance of opportunistic pathogens, such as *Providencia spp.,* have been shown to mediate activation of ISC mitosis in histone demethylase KDM5-deficient flies. KDM5 deficiency causes aberrant immune activation and abnormal social behavior in flies that can be rescued by *L. plantarum* administration [127]. On the contrary, unrestricted growth of *L. plantarum* produces excessive lactate, an essential metabolite that triggers ROS production by the NADPH oxidase Nox. ROS, in turn, induce increased ISC proliferation [128]. Thus, the physiological load of *Lactobacillus spp*. is crucial for maintaining epithelial homeostasis in a lactate-dependent manner. Several studies have reported that microbiota-derived metabolites, such as the previously described SCFAs and bacteriocins, control pathogenic intestinal infection. Bacteriocins are released predominantly by lactic acid-producing bacteria [129] and inhibit peptidoglycan synthesis through binding on the Lipid II of the pathogen’s cell wall, leading to its death [130]. Genetic material within pathogens [131] and their cytoplasmic 16S rRNA are also targeted by bacteriocins to combat infection [132]. This triggers a violent response in the intestine, with aberrant EC loss that greatly increases ISC proliferation [51].

## 5. Conclusions

Nutrient availability is a critical modulator of intestinal microbiota and host metabolic homeostasis. In *Drosophila* and other animals, including mammals, microbiota can ensure or disrupt optimal host nutrient utilization and trigger host immune function. Commensal-induced low damage levels trigger immunity, which modulates basal ISC activity mostly through the JNK and Jak/STAT pathways. Despite the plethora of studies on the impact of different dietary molecules on ISC activity and the impact of microbiota on intestinal renewal, the mechanisms that directly link microbiota-derived metabolites and ISC function remain to be identified. The long application record of *Drosophila* as a successful alternative to mammalian models, makes it a promising system for unsheathing the complex host-microbiome interactions, which are most likely conserved across the animal kingdom.

Although the *Drosophila* gut shares many similarities with the human gut, they also have differences. For example, although the fly midgut offers a unique model to bypass mammalian microbiome complexity, it cannot be humanized, i.e., to be colonized with obligate anaerobic microorganisms. Despite the extent to which it matches the human condition, the real potential of this model is the discovery of vital biological mechanisms linking the microbiome, host nutrition and intestinal function, and homeostasis. Importantly, these findings will also provide important links for the effect of microbiota in immunity, fly physiology and behavior. For example, age-related microbial changes have been correlated with gut inflammation [63,133], whereas the effect of gut microbiota on several aspects of the host behavior, including appetite [57], directed movement, oviposition choice, food choice [134,135], social interactions [127] and mate choice [136] has been the focus of many studies. Future work will elucidate these mechanistic links and will provide useful knowledge for identifying novel treatment for metabolic and ISC-related diseases promoted by bacterial imbalance, such as colon cancer.

## Figures and Tables

**Figure 1 metabolites-12-00169-f001:**
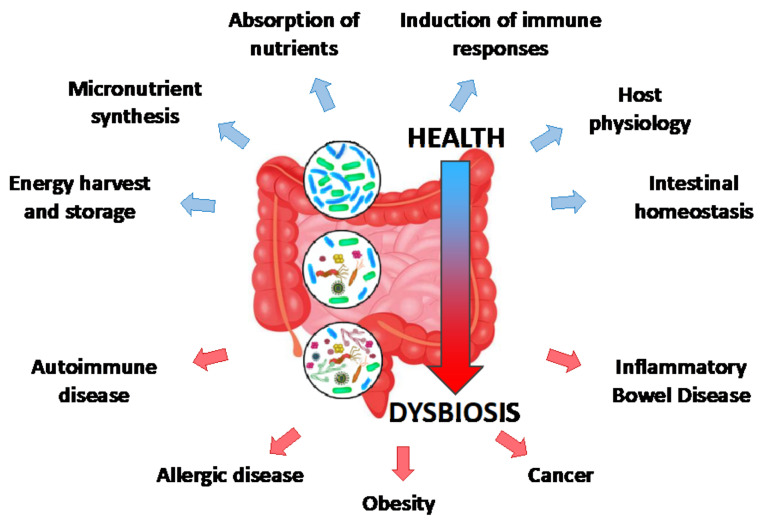
Schematic representation of the transition from healthy microbiota to dysbiosis and its role in human health. Healthy microbiota has a central role in intestinal function, immune response, host physiology and overall fitness. The predominant species of the human intestinal microbiota include *Acetobacter pomorum*, *A. tropicalis*, *Lactobacillus brevis*, *L. fructivorans* and *L. plantarum*. Reduced abundance of these species and concomitant gradual increase of opportunistic pathogens leads to dysbiosis, a condition that predisposes to disease.

**Figure 2 metabolites-12-00169-f002:**
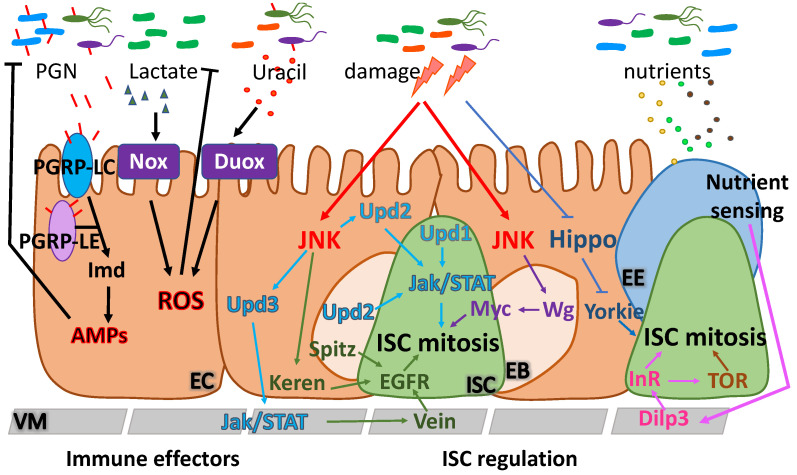
Homeostasis of the *Drosophila* intestinal epithelium is mediated by ISCs that respond to microbiota and damaging agents. Commensal and pathogenic bacteria are sensed by the intestinal epithelium and induce innate immune effectors including ROS and AMPs. Two ROS-producing enzymes, the NADPH oxidases, Duox and Nox, are activated in response to pathogens and commensals, respectively. Duox produces ROS in response to pathogen-derived uracil, whereas Nox produces ROS in response to commensal-derived lactate. The innate immunity Imd pathway controls ROS-resistant microbes. Recognition of the bacterially-derived peptidoglycan (PGN) by the transmembrane receptor PGRP-LC and by the intracellular receptor PGRP-LE leads to Imd activation and subsequent secretion of AMPs. Other PGRPs (PGRP-LB and PGRP-SC) are enzymatic and negatively regulate Imd. Intestinal homeostasis is mediated by ISC mitosis and subsequent differentiation. ISC activity is tightly regulated by evolutionarily conserved growth control signaling mechanisms, including the Hippo, IIS and TOR pathways. ISC proliferation is also regulated by secreted IL-6-like cytokines (Upd1, Upd2, Upd3) that trigger the Jak/STAT pathway. EC damage caused by pathogens, drugs and ROS induces the JNK pathway and elevates Upd3 in ECs, which, in turn, triggers Jak/STAT activation in the VM and the release of EGFs (Vein, Keren and Spitz) by the ISC niche. EGFs activate their receptor EGFR in the ISCs and promote ISC proliferation and tissue regeneration. The Wg morphogen is also induced in EBs and promotes ISC proliferation via cMyc in the infected midguts. EEs sense nutrients and serve as a link between the diet-stimulated Dilp3 expression in the VM, which controls ISC proliferation. (ROS: reactive oxygen species; AMPs: antimicrobial peptides; Duox: Dual oxidase; PGN: peptidoglycan; PGRP: peptidoglycan recognition proteins; InR: Insulin receptor; IIS: insulin/IGF-1 signaling; TOR: target of rapamycin; Upd1,2,3: Unpaired1,2,3; JNK: Jun kinase; EGF: epidermal growth factor; EGFR: epidermal growth factor receptor; Wg: Wingless; Dilp3: *Drosophila* Insulin-like peptide 3; VM: visceral muscle; ISC: intestinal stem cell; EB: enteroblast; EC: enterocyte; EE: enteroendocrine cell).

**Figure 3 metabolites-12-00169-f003:**
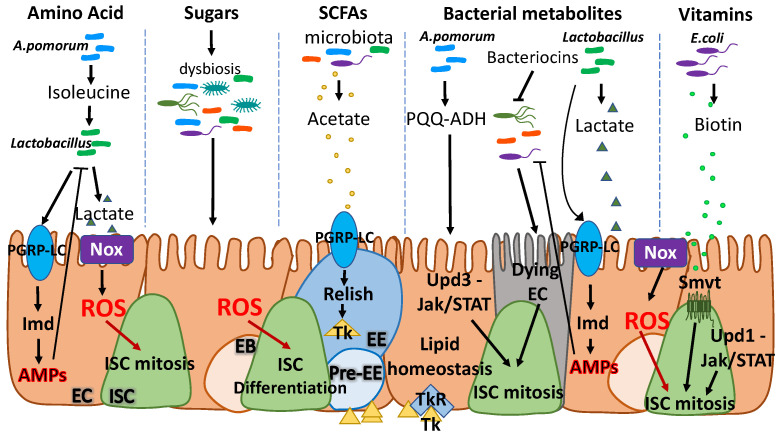
Bacterially-derived micronutrients control *Drosophila* ISC activity. *A. pomorum*-derived isoleucine promotes *L. plantarum* growth, which produces lactate. Lactate is essential for *A. pomorum* growth and induces immune responses in the epithelium to maintain basal ISC numbers. Excess sugar promotes dysbiosis and elevated ROS, which downregulate the Jak/STAT pathway and promote ISC differentiation. The microbiota-derived SCFAs are important regulators of ISC homeostasis. EEs recognize the SCFA acetate and respond by secreting Tk. Tk controls lipid metabolism in ECs to maintain ISC proliferation. Other microbiota-derived metabolites, including the PQQ-ADH, bacteriocins and lactate are important regulators of ISC proliferation. *A. pomorum* PQQ-ADH regulates metabolism, ISC activity and epithelial regeneration via JNK-induced Jak/STAT pathway activation, whereas bacteriocins control pathogenic midgut infection causing EC loss. Lost ECs are replenished via ISC proliferation and differentiation. Moreover, excess lactate produced by *L. plantarum* triggers ROS via Nox, which increases the number of ISCs. Microbiota-produced biotin is absorbed by the ISCs via the Smvt transporter and regulates ISC proliferation in parallel to the Upd1-Jak/STAT pathway. (SCFAs: short chain fatty acids; Tk: Tachykinin; PQQ-ADH: pyrroloquinoline quinone-dependent alcohol dehydrogenase; ISC: intestinal stem cell; EB: enteroblast; EC: enterocyte; EE: enteroendocrine cell).

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
