# Peer review of "How Gut Microbes Nurture Intestinal Stem Cells: A Drosophila Perspective"

_metabolites, 2022, doi:10.3390/metabo12020169_

Round 1
Reviewer 1 Report
The authors reviewed the recent studies about how gut microbes shape intestinal stem cells using Drosophila model. The review is comprehensive, well organized and generally well written. I have only minor suggestions.
- line 92. such us -> such as
- line 141. InR pathway -> IIS pathway like line 195 since there is no term like InR pathway.
- line 186-187. the presence of gut bacteria during the first week of the fly’s adult life impinges positively on longevity [61]. There are numerous references showing that axenic flies outlive conventional flies (see the paper by Min group, 10.18632/aging.102073).
- line 187-188. Life span extension is favored by conditions preserving ISCs and intestinal homeostasis [62]. Better to mention that the relationship between lifespan modulating dietary restriction and ISC proliferation (see 10.1371/journal.pgen.1007777)
- line 253-268. Better to mention the molecular mechanism of ISC modulation by glucose-HBP pathway (see https://doi.org/10.1016/j.devcel.2018.08.011)
Author Response
We thank the reviewer for the helpful criticism. Please see below specific responses to the reviewer’s comments.
- The typo is corrected.
- We have used the term IIS whenever we are referring to the insulin pathway (lines 138, 215, legend of Figure 2).
- We have rewritten the relevant paragraph (lines 192-210) to include the recommended reference (now ref. 68) and additional references to underline the controversial results.
- We have discussed the recommended reference (now ref. 69) in the same paragraph as in point 3.
- We have added the recommended reference (now ref. 84) and discussed the paper (lines 263-269).
Reviewer 2 Report
In this minireview ‚How gut microbes nurture intestinal stem cells: a Drosophila perspective’ Neophytou and Pitsouli review the current literature on the interplay between intestinal microbiota and resident stem cells. The review is precise, very well written and additionally illustrates the main parts with nice figures. It focusses on the most important topics done in the field and provides an exhaustive overview of the topic.
I congratulate the authors for this manuscript, which is why I only have some remarks that might punctually improve the manuscript:
- An upcoming topic in the midgut after the Chen/St Johnston paper in 2018, is the mammal like order of SJ/AJ in contrast to other Drosophila epithelia. The authors may consider to include their findings as it reflects an important barrier against microbiota.
- In the last phrases of 4.3, data showing intestinal plasticity upon mating including fatty acid uptake from the Aliaga/Domniguez/Reiff lab incl. diet related PPARy/Eip75B (Reiff 2015 elife/ Ahmed, 2020, nature, Zipper 2020 elife) could be discussed. Fitting the present speculation, these connections between metabolite uptake/adaptation, circulating hormones and appetite in the last paragraph.
- Font size in all three figures should be increased, when printed A4/letter size, font is too tiny.
Author Response
We wish to thank the reviewer for the helpful criticism. Please see below specific responses to the reviewer’s comments.
- We have included (now ref. 33) and discussed the findings (lines 100-106) of the recommended paper.
- We have discussed the role of JH in ISC proliferation (ref. 103) in Section 4.3 of the revised manuscript (lines 311-314) and we have also included the role of 20HE in midgut homeostasis (ref. 122) in the cholesterol section 4.5 of the review (lines 360-364).
- We have increased the font size of the text in all the figures.
Reviewer 3 Report
The review article written by Constantina Neophytou, et al. summarizes the development of gut microbes in Drosophila animal. The manuscript was very exciting to be read and presents interesting about microbes in Drosophila.
This paper shape and outline can be interesting but the Authors should explain in more points with gut microbiota. The manuscript possesses novelty and therefore it has potential to interest the readers from gut microbiota.
I affirm its acceptance for publication in metabolites with miner revision. However, the following comments must be answered by the authors prior to publication.
Figure 1 looks the human gut images. This is not Drosophila gut. This figure is not related to specific review. Author should revise it.
Most of the paragraph are too length to read and understand. Author should split it.
This topic of conclusions should be developed. What are the hallmarks of gut microbes in Drosophila?.
Author Response
We would like to thank the reviewer for the criticism. Please see below specific responses to the reviewer’s comments.
- Indeed, Figure 1 outlines the human gut-microbe interaction. We explicitly mention that in the figure legend (now in its title too, line 42). We wish to maintain this figure as is, because we believe it is a useful introduction to the theme.
- We have split the long paragraphs, as per reviewer’s recommendation, to facilitate reading.
- We have expanded the “conclusions” section of the review to include the pros and cons of studying host-microbiota interactions in flies and we comment on the effects of gut microbiota on fly health explicitly.